# Generation and Immunogenicity of Virus-like Particles Based on the Capsid Protein of a Chinese Epidemic Strain of Feline Panleukopenia Virus

**DOI:** 10.3390/vetsci12050503

**Published:** 2025-05-20

**Authors:** Erkai Feng, Guoliang Luo, Chunxia Wang, Wei Liu, Ruxun Yan, Xue Bai, Yuening Cheng

**Affiliations:** 1Key Laboratory of Economic Animal Diseases, Ministry of Agriculture, Institute of Special Animal and Plant Science, Chinese Academy of Agriculture Science, Changchun 130111, China; fengerkai@caas.cn (E.F.); luoguoliang@caas.cn (G.L.); wangchunxia02@caas.cn (C.W.); 2Changchun Xinuo BioTechnology Co., Ltd., Changchun 130111, China; yfsb@ccsino.com.cn (W.L.); zlglb@ccsino.com.cn (R.Y.)

**Keywords:** feline panleukopenia, feline panleukopenia virus, baculovirus expression, virus-like particles, vaccine

## Abstract

Feline panleukopenia (FPL) is a severe, highly contagious viral disease affecting nearly all felids. Given its high morbidity and mortality rates, prevention through vaccination is critical. In this study, we developed FPLV-like particles (VLPs) incorporating the capsid protein VP2 using a baculovirus expression vector system (BEVS). These VLPs were formulated with Seppic adjuvant to create a candidate vaccine. Immunization with this vaccine elicited a strong antibody response, demonstrated by a hemagglutination titer of 1:2^16^. Furthermore, vaccinated cats achieved 100% protection against challenge with a virulent FPLV variant (Ala91Ser/Ile101Thr). These findings highlight the potential of FPLV-VLP-based vaccines as a safe and effective strategy for preventing FPLV infection in cats.

## 1. Introduction

Feline panleukopenia (FPL), predominantly caused by feline panleukopenia virus [1], is the earliest documented parvovirus infections in felids [2]. FPL has a nearly ubiquitous prevalence across global feline populations with a morbidity rate exceeding 90% in naïve populations, while mortality escalates to 50–90% in kittens [3]. Vaccine administration remains the cost-effective approach for preventing and controlling this debilitating condition [4,5].

However, in China, vaccinating cats against FPLV is not accepted by every cat owner, either because of the high price or low awareness of vaccination. For cat owners who want to vaccinate their cats, they have no choice but to opt for the Fel-O-Vax^®^ PCT, which is the only foreign killed combination vaccine that is approved for sale in China so far. The lack of vaccines, especially localized vaccines, means that the healthy and stable development of China’s growing pet industry cannot be guaranteed.

The advent of virus-like particles (VLPs) has redefined vaccinology by merging structural virology with immunology [6]. The term VLPs refers to particles that self-assemble with one or more viral structural proteins [7]. Unlike traditional vaccines, VLPs morphologically resemble viruses but without the replicative risks [8]. Moreover, they also differ from other subunit vaccines in that they have strong immunogenicity due to the presentation of repetitive antigenic epitopes in the immune system [9,10].

Multiple expression systems have been employed for VLP production, and there is a degree of flexibility in the manufacturing conditions necessary for large-scale production of VLPs [8]. Of these techniques, the baculovirus expression system (BES) has emerged as a cornerstone technology for producing structurally authentic VLPs, owing to its unique capacity to preserve post-translational modifications (PTMs) (glycosylation) and be scaled cost-effectively [8,11].

This study pioneers the use of the BES to generate VLPs expressing the VP2 proteins of a FPLV Chinese epidemic strain (Ala91Ser) of feline panleukopenia virus (FPLV-VP2-VLPs), and an investigation to characterize and assess the immunogenicity of these FPLV-VLPs was conducted. The results gained from this research could offer fundamental data for the development of innovative FPLV vaccines. These findings position FPLV-VLPs as a cornerstone for pan-parvovirus vaccine development, with translational potential extending to canine parvovirus (CPV) and porcine parvovirus (PPV).

## 2. Materials and Methods

### 2.1. Virus and Cells

Feline panleukopenia virus (named FPLV-CC19-02), used in this study, was previously isolated from a cat with severe diarrhea in Changchun City, Jilin Provine, harboring an Ala91Ser substitution in the VP2 protein; this virus has been widely circulated in cats in China since 2017 [12,13]. The complete genome sequence of FPLV-CC19-02 is publicly available (accession no. OR921195.1). *Spodoptera frugiperda* (Sf9) was cultured in Sf9-900 II SFM medium (Gibco, Waltham, MA, USA) at 28 °C. Cellfectin™ II transfection reagent was purchased from Gibco company (Beijing, China), and a BacPAK Baculovirus Rapid Titer Kit was obtained from Takara company (Dalian, China). The anti-CPV-2c-VP2 monoclonal antibody (clone 5B18) was homemade in our laboratory.

### 2.2. Generation of Recombinant Baculoviruses

The sequence encoding the VP2 protein from FPLV-CC19-02 (OP471917.1) was optimized for *Spodoptera frugiperda* 9 (Sf9) cells and was synthesized by Nanjing Zoonbio Biotech Co., Ltd. (Nanjing, Jiangsu, China) based on specific primers: VP2-F, 5′-GATTATTCAACCGTCCCACCATCGGGCGCGGATCCGCCACCATGCTGCTGGTGAACAGAGCCACCAG-3′, and VP2-R, 5′-GCTGATTATGATCCTCTAGTACTTCTCGACAAGCTTTTAGGCGTAGTCAGGCACGTCGTAGGGGTAGTA-3′. The optimized sequence was specifically designed to enhance its expression in Sf9 cells.

The synthesized target gene was identified by PCR and subcloned into the donor plasmid *pFastBac* by homologous recombination. After synthesis was confirmed by double-restriction enzyme digestion (*BamH* I and *Hind* III) and PCR identification, the recombinant plasmids (*pFastBac-FPLV-VP2*) were transformed into *E. coli* DH10Bac-competent cells to produce the rBacmid-FPLV-VP2. Blue–white screening was employed to detect recombinant bacmid DNA colonies, which were further identified by PCR amplification using the M13 primers. The recombinant bacmids were isolated and purified using the Endo Free Plasmid Maxi kit (QIAGEN, Hilden, Germany), and the process was performed in accordance with the manufacturer’s protocol.

### 2.3. Acquisition of Recombinant Baculoviruses

The Sf9 cell suspension (5 × 10^6^ cells/mL) was seeded into 6-well cell plates (2 mL/well) and cultured at 27 °C for 12 h. Then, 4 μg of bacmid-FPLV-VP2 and 8 μL of Cellfectin™ II transfection reagent (Gibco, Baltimore, MD, USA) were added to 100 μL of Grace’s medium (no serum, no antibiotic), respectively, and incubated for 30 min at 37 °C in incubators. Subsequently, the two reagents were gently combined and incubated for a further 30 min at room temperature. Following this, the mixture was dispensed onto the Sf9 cells and added to the cell culture plate. The cells were then incubated at 28 °C for 4 h before being replaced with 2 mL of Sf-900 II SFM medium. After two serially passaged, virus-infected Sf9 cells showed typical phenotypic symptoms of viral infection, like an increase in cell diameter, the cells were detached from the plate.

### 2.4. Indirect Immunofluorescence Assay (IFA) Analysis of Recombinant Baculoviruses

The suspension of Sf9 cells (2 × 10^6^ cells/mL) was seeded in 6-well cell plates and infected with the four-generation recombinant baculoviruses. For the negative control, uninfected Sf9 cells were used. Two days after infection, the culture medium was removed, and the Sf9 cells were washed three times with PBST, each for 5 min. Later on, the cells were fixed using 4% paraformaldehyde for 45 min at room temperature. After three more washes with PBST, 0.5% Triton X-100 was added to the cells to facilitate permeabilization. Following this step, the solution was discarded, and the cells were washed three additional times. Then, the cells were blocked with 5% skimmed milk for 60 min at 37 °C. After three more washes, the cells were incubated with a homemade mouse anti-CPV-2c monoclonal antibody (clone 5B18), diluted at a ratio of 1:200, for 1 h at 37 °C. Following this, the cells were washed three times and then incubated with a fluorescein isothiocyanate (FITC)-labeled goat anti-mouse IgG (H + L) antibody diluted at a ratio of 1:2000 for 1 h at 37 °C. Finally, the cells were washed thoroughly three times and observed under a fluorescence microscope (Leica AF 6000, Wetzlar, Germany).

### 2.5. Generation and Purification of FPLV-VP2-VLPs

To generate the FPLV-VP2-VLPs, 100 mL of Sf9 cell suspension (5 × 10^8^ cells) was infected with four-generation recombinant baculoviruses at a multiplicity of infection (MOI) of 1 and incubated at 28 °C for 4–5 days. Subsequently, the Sf9 cells were harvested through centrifugation at 8000× *g* for 20 min using a rotator-type F-34-6-38 centrifuge from Eppendorf, Germany. The collected cells were lysed with 25 mM NaHCO_3_ at 4 °C for 2 h, and the resulting supernatant, containing the FPLV-VP2-VLPs, was further centrifuged. To analyze the expression of the VP2 protein, SDS-PAGE and Western blotting techniques were employed, utilizing a homemade anti-CPV-2c monoclonal antibody (clone 5B18) as the primary antibody.

### 2.6. Analysis of the Antigenicity of FPLV-VP2-VLPs

The antigenicity of the FPLV-VP2-VLP protein, designated as P4, was assessed using rapid immunochromatographic test strips developed by SHANGHAI QUICK-ING Biotech Co., Ltd. (Shanghai, China), which are specifically used for detecting feline panleukopenia virus antigens.

### 2.7. Hemagglutinin Activity Analysis of FPLV-VP2-VLPs

The hemagglutinin activity of the FPLV-VP2-VLPs was evaluated through the hemagglutinin assay (HA) [14] utilizing a 96-well ‘V’-bottom microtiter plate. To begin, 25 μL of 0.1 M PBS (pH 6.4) was evenly distributed in well A and subsequently across the rows (1–4) and columns (A–G) using a 10–100 μL variable-volume multichannel micro-pipette. Then, 25 FPLV-VP2-VLPs were introduced into the first well of the microtiter plate, establishing one biological replicate and control group. Then, they were serially diluted from 1:2^1^ to 1:2^20^ across the first column. Afterward, 25 μL of PBS was added to all active wells. Thereafter, 50 μL of 1% porcine red blood cell (RBC) suspension was introduced. The plate was gently shaken at 150 rpm for 2 min to ensure proper mixing of the reactants. Following this, the plate was incubated at 4 °C for 45 min, and the formation of serrated-edged mats or bottoms was recorded, indicating negative and positive results, respectively. The titer was calculated as the reciprocal of the last well with agglutination.

### 2.8. Transmission Electron Microscopy

The morphology of the FPLV-VP2-VLP protein (P4) was examined under transmission electron microscopy (TEM) following negative staining, as described previously by Gao et al. [15]. Specifically, 1 mL of the FPLV-VP2-VLP protein was combined with 20 μL of CaHPO4 solution in a 1.5 mL Eppendorf tube, thoroughly mixed, and incubated for 10 min at room temperature. Subsequently, the tube was centrifuged at 15,000 r/min for 15 min, and the resulting sediment was dissolved with 15 μL of an EDTA-saturated solution to form a droplet. This droplet was placed on a copper grid and allowed to stand for 20 min before being stained with 2% phosphotungstic acid staining solution (pH 6.8) for 1 min. Excess staining solution was removed with filter paper, and the virus morphology was then observed through a JEOL 2010 transmission electron microscope (JEOL Ltd., Tokyo, Japan) operated at an acceleration voltage of 100 kV.

### 2.9. Animal Immunization and Challenge

A total of twelve seronegative British shorthair cats, aged between 3 and 4 months, were divided into four groups in a randomized manner: Group I—VLP vaccine (Seppic adjuvant, 9:1); Group II—killed FPLV vaccine (Seppic adjuvant, 9:1); Group III—Fel-O-Vax^®^ PCT; and Group IV—control (minimum essential medium (MEM)). Each cat was initially administered with a single dose of 1.0 mL of each sample. Blood samples were systematically collected from each cat at various time points post-vaccination, including on days 0, 7, 14, 21, 24, 28, 35, 42, and 50. To determine the serum antibody titers against FPLV, a hemagglutination inhibition assay (HI) was employed.

Three weeks after vaccination, all cats were orally challenged with 5 mL of FPLV-CC19-02 strain cell culture (HA = 2^10^, TCID_50_ = 10^6.5^/mL). Following this challenge, continuous observation and recording of the cats’ performances were conducted for a period of 10 days post-challenge (dpc). This included monitoring their mental state, food and water intake, excrement consistency, and rectal temperature. These observations were used to assess the immune efficacy of the vaccines and to evaluate the protection they provided and the generation of an effective immune response in the cats. This monitoring process helped us to evaluate the vaccines’ effectiveness in providing strong immunological defense against the FPLV-CC19-02 strain.

To evaluate the safety of the vaccine, muscle tissue was collected from the vaccine inoculation site of cats in the experimental and control groups, fixed in 10% paraformaldehyde, processed via paraffin-embedding, mounted on slides, and stained with hematoxylin and eosin (HE). Finally, it was analyzed through histopathology analysis. Before collecting the samples, all cats, either dying or in good health, were euthanized via an intravenous injection of propofol (0.6 mg/kg, Jiabo Co., Ltd., Guangdong, China) and potassium chloride solution (100 mg/kg, MACKLIN Co., Ltd., Shanghai, China).

### 2.10. Statistical Analysis

The analysis of the collected data was conducted using GraphPad Prism Software version 9.5 (GraphPad Software, San Diego, CA, USA). The results were presented as the mean values accompanied by their corresponding standard deviation (SD). A one-way analysis of variance (ANOVA) was utilized to compare the data among three or more experimental groups.

## 3. Results

### 3.1. Construction and Identification of the Recombinant Bacmid-VP2

A gene fragment, approximately 2000 base pairs (bp) in size, was obtained through overlapping extension PCR amplification (Figure 1A, lanes 1 and 2). Later on, the VP2 gene was subcloned into the donor vector *pFastBac*-I through homologous recombination. This was verified through PCR (Figure 1B, approximately 2200 bp, lanes 1–8) and confirmed by double enzyme digestion (Figure 1C, lanes 2), showing 926 bp and 4671 bp fragments, respectively. Subsequently, *pFastBac*-VP2 was introduced into *E. coli* DH10Bac-competent cells to generate the recombinant bacmid-VP2. The successful amplification of the target band, at approximately 4100 bp (Figure 1D, lanes 3, 5–10), indicated the successful construction of the recombinant bacmid-VP2.

### 3.2. Analysis of FPLV-VP2-VLPs

Based on SDS-PAGE analysis, it was clear that the FPLV-VP2-VLP protein, weighing approximately 70 kDa, was predominantly present in the supernatant of NaHCO_3_-treated infected Sf9 cells (Figure 2A, lane 2). Additionally, immunoblotting analysis indicated that the expression of FPLV-VP2-VLPs was intracellular (Figure 2B, lane 2, arrows).

Further examination through transmission electron microscopy (TEM) revealed the presence of numerous circular and spherical virus-like structures with a diameter range of 20–30 nm, resembling the native FPLV, in the supernatant of the treated cells. These structures closely resemble the naïve FPLV (Figure 2C).

When FPLV-VP2-VLPs were serially diluted (10^0^ to 10^−3^), they could still be detected by the FPLV antigen rapid test immunochromatographic strips, indicating their high antigenicity (Figure 2E). Additionally, the hemagglutination (HA) titer of FPLV-VP2-VLPs (P4) reached a high level of 1:2^14^~1:2^16^ (Figure 2D).

**Figure 1 vetsci-12-00503-f001:**
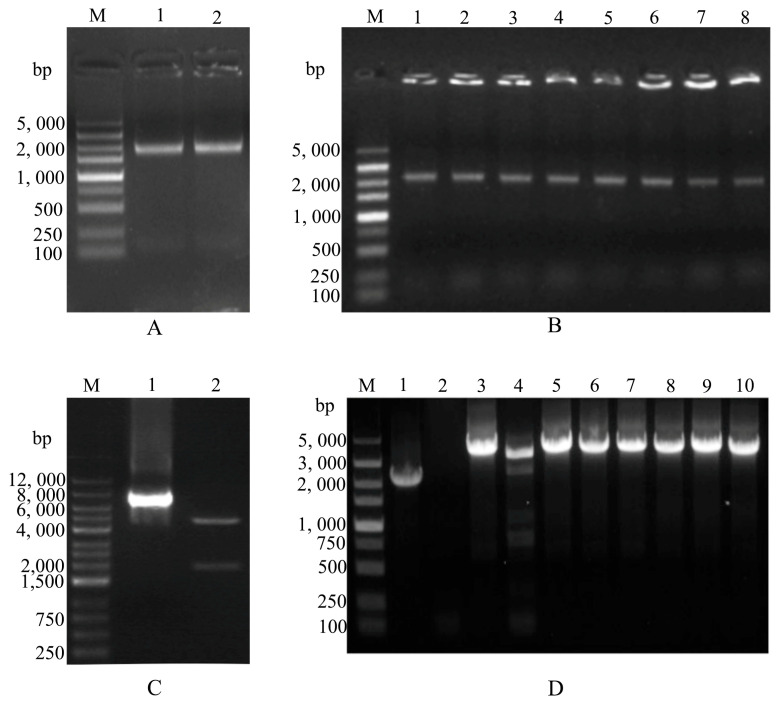
Construction and identification of recombinant bacmid-VP2. (**A**) PCR amplification of the FPLV-VP2 gene fragment. M: DL 5000 marker; lanes 1–2: FPLV-VP2 gene. (**B**) PCR identification of the recombinant transfer plasmid FPLV-VP2. M: DL 5000 marker; lanes 1–8: white colonies on the Luria–Bertani (LB) plate. (**C**) Double-restriction enzyme digestion identification of the recombinant transfer plasmid pFastBac-FPLV-VP2. M: DL 12,000 marker; lane 1: pFastBac-FPLV-VP2; lane 2: pFastBac-FPLV-VP2 digested with restriction endonucleases BamH I and Hind III. (**D**) PCR identification of recombinant bacmid-VP2. M: DL 5000 marker; lane 1: positive control; lane 2: negative control; lane 3 and lanes 5~10: white colonies.

### 3.3. Indirect Immunofluorescence Assay

Sf9 cells infected with the recombinant baculovirus FPLV-VP2-VLP showed an increase in diameter and exhibited high-intensity green fluorescence (Figure 3B,D), whereas the control group cells demonstrated no cytopathic effect (CPE) or visible fluorescence (Figure 3A,C). This observation clearly indicates that the Sf9 cells infected with the recombinant baculovirus correctly expressed the FPLV-VP2 protein.

### 3.4. Change in Hemagglutination Inhibition (HI) Antibody After Vaccination

The hemagglutination inhibition (HI) antibody titer of all cats following vaccination and challenge is illustrated in Figure 4. At 7 days post-vaccination (dpv), two cats in Group II (FPLV inactivated vaccine) exhibited detectable hemagglutination inhibition antibodies (red line/scatter). At 14 dpv, all cats in Groups I (green line/scatter) and II (red line/scatter) showed elevated levels of HI antibodies. By 21 dpv, cats in all groups, except the control group (black line/scatter), had elevated levels of HI antibodies. Following the challenge, the titer of HI antibodies continued to increase in all cats, reaching a new peak value unless they succumbed to the challenge.

**Figure 2 vetsci-12-00503-f002:**
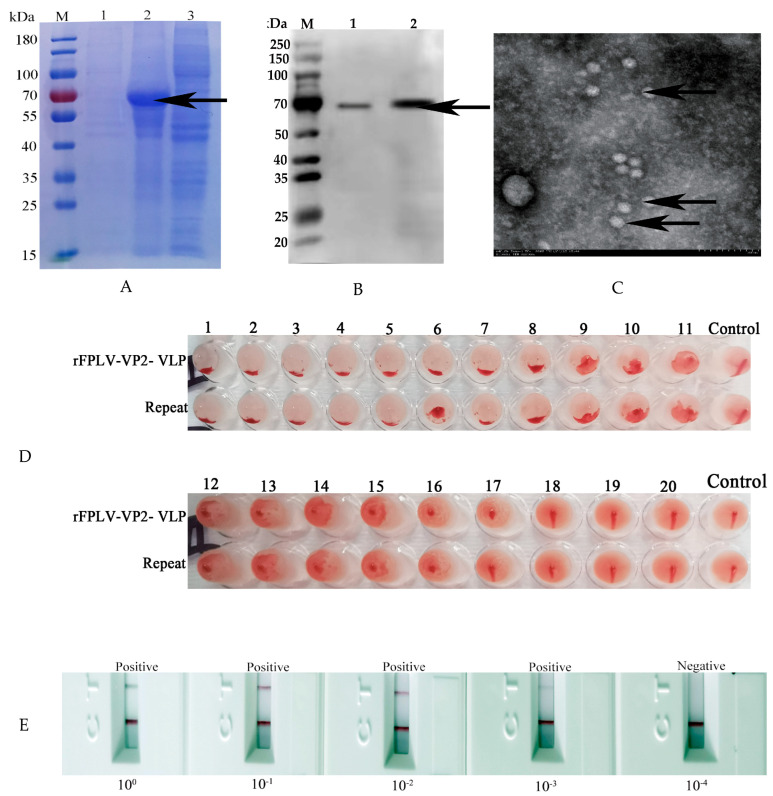
Analysis of FPLV-VP2-VLPs. (**A**) SDS-PAGE analysis of FPLV-VP2-VLPs. M: PageRuler pre-stained protein marker; lane 1: supernatant of NaHCO_3_-treated Sf9 cells (not infected); lane 2: supernatant of NaHCO_3_-treated Sf9 cells (infected); lane 3: cell precipitation from NaHCO_3_-treated Sf9 cells (infected). (**B**) Immunoblotting analysis of FPLV-VP2-VLPs. M: protein marker; 1: cell precipitation of NaHCO_3_-treated Sf9 cells (infected) Original image Appendix A; 2: supernatant of NaHCO_3_-treated Sf9 cells (infected). (**C**) TEM observation of FPLV-VP2-VLPs (4000×). (**D**) Hemagglutination analysis of FPLV-VP2-VLP protein. (**E**) Antigenicity analysis of FPLV-VP2-VLP protein.

### 3.5. Clinical Characteristics of Cats After Challenge

The resulting challenge-protection statistics are presented in Table 1. Both the VLP vaccine and FPLV killed vaccine, prepared with Seppic adjuvant, offered 100% protection to the cats. However, the commercial vaccine (Fel-O-Vax^®^ PCT) provided only partial protection to the cats (33%). All cats in the control group succumbed to the challenge either at 3 or 8 dpc.

The changes in body temperature for cats in all groups after the challenge are depicted in Figure 5. Approximately 67% of the cats vaccinated with the commercial vaccine (Fel-O-Vax^®^ PCT) exhibited hyperpyrexia (>40 °C in average) (Group III, pink line/scatter). The temperature of cats in Groups I (red line/scatter) and II (green line/scatter) float within a normal range. Interestingly, the body temperature of cats in the control group initially decreased before increasing to higher temperatures (>40 °C). Some cats’ body temperatures dropped to lower levels before they passed away (pink line/scatter).

Furthermore, after the virus challenge, cats in Groups III and IV displayed clinical manifestations of feline panleukopenia (FPL). These signs included fever, depression or lethargy, vomiting, diarrhea, unconsciousness, and even fatal outcomes (Table 2). In contrast, the cats in Groups I and II remained asymptomatic.

**Figure 3 vetsci-12-00503-f003:**
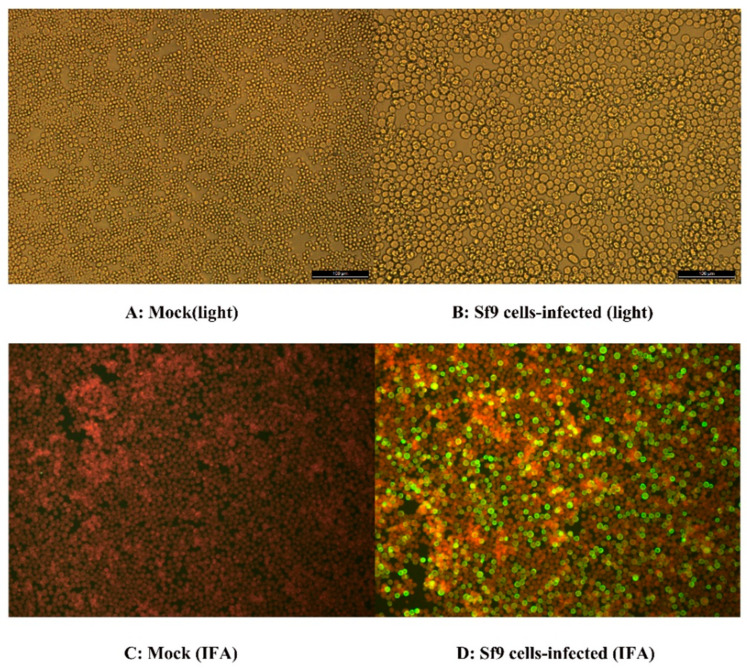
The indirect immunofluorescence assay (IFA) of the recombinant FPLV-VP2. (**A**) Sf9 cells (light); (**B**) Sf9 cells infected with recombinant baculovirus (light); (**C**) Sf9 cells (IFA); (**D**) Sf9 cells infected with recombinant baculovirus (IFA).

**Figure 4 vetsci-12-00503-f004:**
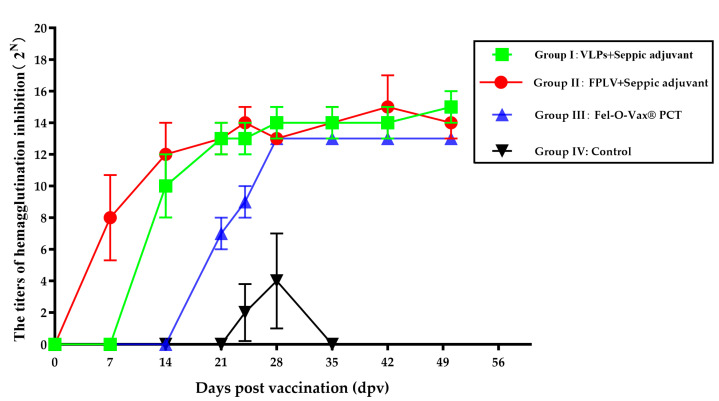
The dynamic response process of all cats (hemagglutination inhibition antibody titer) to vaccination with VLP vaccine and challenge with virulent strain of FPLV.

**Figure 5 vetsci-12-00503-f005:**
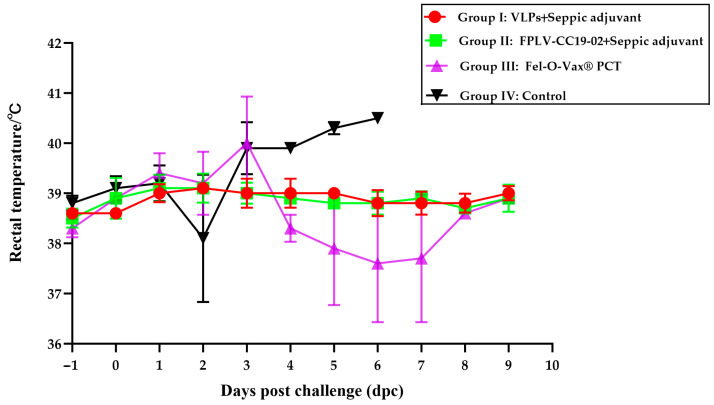
The change in body temperature of cats during the days post-challenge (dpc).

### 3.6. Safety Evaluation of Vaccines

After necropsy, no sarcomas were detected at the injection site (Figure 6A). In both the control group (Figure 6B) and the vaccine group (Figure 6C,D), the muscle cells exhibited a neat and tight arrangement with distinct cell boundaries and a consistent orientation. The horizontal lines of muscle cells were clearly visible, alternating between light and dark hues, and there were no abnormalities in the stroma. Moreover, there were no evident infiltrations of inflammatory cells observed in any of the examined tissue samples.

**Figure 6 vetsci-12-00503-f006:**
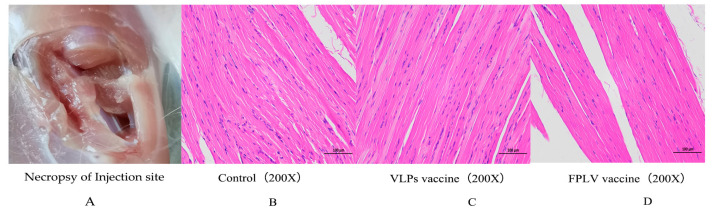
The injection site necropsy and histopathology examination results of muscle tissue from the vaccine inoculation site. (**A**) Necropsy of tissue from the inject site of vaccine. (**B**) Histopathology of tissue from the inject site of cats in the control group. (**C**) Histopathology of tissue from the inject site of cats in VLPs-based vaccine. (**D**) Histopathology of tissue from the inject site of cats in FPLV vaccine.

## 4. Discussion

Feline panleukopenia virus (FPLV) is the major etiological factor of FPL (90–95% of cases), although canine parvovirus variant infections in cats have been occasionally reported (<10% of cases). Vaccination remains a cost-effective measure to prevent and control virus infection in cats. However, in China, the status quo with regard to cat vaccination is just the use of a foreign killed combination vaccine (Fel-O-Vax^®^ PCT), as this is the only vaccine that has been approved for sale so far, with its viral strains collected over 30 years.

In recent years, many FPLV variants carrying a substitution in the capsid protein have emerged in China in different hosts, like Gly299Glu (in giant pandas) [16], Ala300Pro (in dogs) [17,18], and Ala91Ser (in cats and dogs) [12,13,19]. Of these, the Ala91Ser FPLV variant deserves more attention due to its heightened infectivity and pathogenicity [19], and it has become dominant among FPLV clinical isolates in China since 2019 [12,19] with a clinical isolation rate of over 50%, peaking as high as 86% (Appendix A). Meanwhile, concerns about the ability of the traditional vaccine to protect against FPLV variant infection have been raised, and high-safety vaccines are urgently needed to ensure the healthy and stable development of China’s pet industry.

The advantages of virus-like particle (VLP)-based vaccines are apparent, such as their structural resemblance to their parental viruses [9] and their non-infectious nature due to the lack of a viral genome. In this study, we successfully expressed the VP2 protein of FPLV in an insect cell baculovirus expression system (Figure 2A). The expressed VP2 protein correctly folded into VLPs (Figure 2C), which were identified using a monoclonal antibody to the expressed VP2 of canine parvovirus 2c (Figure 2B and Figure 3D) and confirmed via FPLV antigen rapid test immunochromatographic strips (Figure 2E). These results demonstrated the high antigenicity of the expressed protein. Furthermore, the VLPs exhibited enhanced hemagglutination activity, achieving titers of 1:2^14^ to 1:2^16^ (Figure 2D), surpassing the HA titer of the epidemic strain of FPLV, which ranged from 1:2^8^ to 1:2^10^.

FPLV-VP2-VLPs demonstrate robust immunogenicity, eliciting a significant humoral immune response in vaccinated cats. At 14 days post-vaccination (dpv) with the Seppic-adjuvanted VLP vaccine, cats exhibited a high hemagglutination inhibition (HI) antibody titer of 1:2^12^, consistent with prior research [20]. By 21 dpv, all vaccinated cats achieved HI titers exceeding 2^10^, indicating robust and sustained immunity. Notably, the whole-virus inactivated vaccine also induced a positive immune response, with some cats showing detectable HI antibodies as early as 7 dpv. These findings suggest that VLP-based vaccines are comparable in efficacy to traditional vaccines. In contrast, cats administered the commercial Fel-O-Vax^®^ PCT vaccine demonstrated delayed antibody detection, with HI titers remaining below 2^8^ until 21 dpv (Figure 4, blue line/scatter).

The level of antibodies generated by vaccines is a crucial determinant of protective efficacy against viral challenges. In this study, both the VLP-based vaccine (Seppic adjuvant) and whole-virus inactivated vaccine elicited significant protection against virulent FPLV challenge, as evidenced by 100% survival rates in the vaccinated cohorts. In contrast, the commercial vaccine demonstrated only 33% protection, potentially attributable to suboptimal immunization protocols (single-dose administration). These findings align with broader vaccine research indicating that antibody titers ≥ 1:64 are generally required for feline panleukopenia protection, while suboptimal dosing strategies may compromise immune response.

The safety of companion animal vaccinations has been a topic of significant debate over the past decade, particularly regarding the association between vaccines and sarcoma development in cats [21,22]. To evaluate the safety of our FPLV-VP2-VLP vaccine, we performed histopathological analysis of muscle tissue samples collected from the inoculation sites of vaccinated cats. Notably, no sarcomas or pathological alterations were observed in any animals (Figure 6), confirming the absence of vaccine-associated sarcoma formation. Additionally, all vaccinated cats maintained a clinically normal body temperature throughout the study period. Collectively, these findings demonstrate the greater safety profile of the VLP-based vaccine.

## 5. Conclusions

In conclusion, a pioneering pet vaccine against FPL has been developed in this study, demonstrating greater effectiveness and an improved safety profile. In future applications, the FPLV-VLP vaccine can serve as a standalone vaccine or be combined with other vaccines for effective immunization of cats against FPL, providing comprehensive phylaxis.

## Figures and Tables

**Table 1 vetsci-12-00503-t001:** Challenge protection results obtained for different experimental groups.

Groups	Vaccine	Number of Cats	Number of Diseased Cats	Protection Rate
Group I	FPLV-VLP vaccine (Seppic adjuvant)	3	0	3/3 (100%)
Group II	FPLV killed vaccine (Seppic adjuvant)	3	0	3/3 (100%)
Group III	Commercial vaccine (Fel-O-Vax^®^ PCT)	3	2	1/3 (33%)
Group Ⅳ	Minimum essential medium	3	3	0/3 (0%)

**Table 2 vetsci-12-00503-t002:** Statistics of clinical manifestations in experimental cats after virus challenge.

Groups	Vaccine Type	Number of Cats		N Days Post-Challenge (dpc)									
			−1	0	1	2	3	4	5	6	7	8	9
Group I	FPLV-VLP vaccine (Seppic adjuvant)	1	0	0	0	0	0	0	0	0	0	0	0
		2	0	0	0	0	0	0	0	0	0	0	0
		3	0	0	0	0	0	0	0	0	0	0	0
Group II	FPLV killed vaccine (Seppic adjuvant)	1	0	0	0	0	0	0	0	0	0	0	0
		2	0	0	0	0	0	0	0	0	0	0	0
		3	0	0	0	0	0	0	0	0	0	0	0
Group III	Commercial vaccine (Fel-O-Vax^®^ PCT)	1	0	0	0	0	0	0	0	0	0	0	0
		2	0	0	0	0	0	1	1	2	2	3	
		3	0	0	0	0	0	0	1	1	1	1	3
Group IV	Control Group (MEM)	1	0	0	0	0	0	0	1	1	2	3	
		2	0	0	0	1	3						
		3	0	0	0	0	0	1	1	1	2	3	

Note: “0”—no symptom; “1”—depression or languish or slight fever; “2”—fever more than 40 °C or unconscious; “3”—died.

## Data Availability

All data are contained within this article and its Appendix A.

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
