# Peer review of "Generation and Immunogenicity of Virus-like Particles Based on the Capsid Protein of a Chinese Epidemic Strain of Feline Panleukopenia Virus"

_vetsci, 2025, doi:10.3390/vetsci12050503_

Round 1
Reviewer 1 Report
Comments and Suggestions for Authors
It is important to answer two questions in this manuscript.
First, when the 37% turnitin report of the article was examined, it was seen that there was a very high similarity with several articles. Moreover, no article with this similarity is included in the reference list. You can see below theses:
. Arch Virol. 2020 Sep;165(9):2065-2071. doi: 10.1007/s00705-020-04703-6. Epub 2020 Jul 1: Generation and immunogenicity of virus-like particles based on mink enteritis virus capsid protein VP2 expressed in Sf9 cells
. Veterinary Microbiology: Volume 248, September 2020, 108795. https://doi.org/10.1016/j.vetmic.2020.108795: The immunogenicity of the virus-like particles derived from the VP2 protein of porcine parvovirus
. Vaccines 2021, 9(1), 53; https://doi.org/10.3390/vaccines9010053
Secondly; when the doi of this article received from research square in 2023 is examined, there is a list of 11 authors, while there are 6 author names in the format entered in Veterinary Science.
The reason for these situations should be explained.
The parts of the article that require correction are listed below.
It is known that feline panleukopenia viruses as well as canine parvovirus type 2 variants can cause infection in cats. Therefore, the limitations of the article should be explained. The success of this VLP vaccine, which was prepared only for a variant in China, in infections against other strains should not be questioned. In this context, the findings obtained are specific only for this variant.
It is important to include in the discussion how many cats are affected by this new variant. The reason that led researchers to prepare a vaccine against this new variant should be explained.
In Figure 6, only the situation in the muscles at the injection site was evaluated. Why was necropsy not performed on other tissues? Or if it was performed, why were the results not shared? It is important to minimize the effects of Parvoviruses, which cause active infection in the gastrointestinal tract, on this system with vaccination. In particular, the immunohistochemistry results between the unvaccinated group and the vaccinated group that received oral challenge virus are important to provide information about the success of this vaccine. It is recommended that this information be added to the discussion.
Figure 2B is an unclear image. The kilodalton size of the VLP bands is not specified.

Author Response
Response to Reviewer 1 Comments
|
||
1. Summary |
|
|
Thank you very much for taking the time to review this manuscript” Generation and immunogenicity of virus-like particles based on Capsid Protein of a Chinese Epidemic Strain of Feline Panleukopenia Virus (vetsci-3574053)”. Those comments are all valuable and very helpful for revising and improving my paper. We have studied comments carefully and have made correction which we hope meet with approval. Please find the detailed responses below and the corresponding corrections highlighted in green in the re-submitted files. |
||
2. Point-by-point response to Comments and Suggestions for Authors |
||
Comments 1: [First, when the 37% turn it in report of the article was examined, it was seen that there was a very high similarity with several articles. Moreover, no article with this similarity is included in the reference list. You can see below theses: Arch Virol. 2020 Sep;165(9):2065-2071. doi: 10.1007/s00705-020-04703-6. Epub 2020 Jul 1: Generation and immunogenicity of virus-like particles based on mink enteritis virus capsid protein VP2 expressed in Sf9 cells. Veterinary Microbiology: Volume 248, September 2020, 108795. https://doi.org/10.1016/j.vetmic.2020.108795:The immunogenicity of the virus-like particles derived from the VP2 protein of porcine parvovirus. Vaccines 2021, 9(1), 53; https://doi.org/10.3390/vaccines9010053] |
||
Response 1: Dear reviewer, thank you for pointing this out. 37%, indeed, is very high similarity between my manuscript and several articles. However, after I deeply checked the articles that examined, we found that the most similarity are from a preprinted article (10.21203/rs.3.rs-3569093/v1), which was a previous version of this manuscript (Generation and immunogenicity of virus-like particles based on Capsid Protein of a Chinese Epidemic Strain of Feline Panleukopenia Virus), so it is no wonder shows higher similarity between them. As to the above two articles, we did not know which words or sentences in manuscript (vetsci-3574053) show high similarity with the above two articles, so, I beg the reviewer 1 to tell me the detailed information about the similarity. In addition, some similarity between the manuscript and the temperate of Veterinary Science have changed, details can be seen in |
||
Comments 2: [Secondly; when the doi of this article received from research square in 2023 is examined, there is a list of 11 authors, while there are 6 author names in the format entered in Veterinary Science. The reason for these situations should be explained.] |
||
Response 2: Dear reviewer, thank you for pointing this out. A manuscript, named “Baculovirus Expression of Feline Panleukopenia Virus Capsid Protein and Induces Robust Immune Responses in Cats”, was pre-printed on research square (2023.11.30), but it has not been peer reviewed by a journal. In the manuscript, there is indeed a list of 11 authors. The reason for why there are 6 author names in the manuscript that submitted to Veterinary is that some students had graduated from the school, such as Manping Yan, Jinyuan Shang, and Zhigang Cao. The reason for removing the professor Cheng Shipeng and Zhang Xiaohao from the manuscript is that they had retired from their work unit since 2024, respectively. According to the above reasons, these authors are no longer appropriate for the manuscript. So, we removed their name from the manuscript after obtaining their consent. The reason for adding the author (Ran Ruxun) to the manuscript is that he helps me a lot in drawing the picture and polished the manuscript. As to the reason for adding the Professor ’Bai xue’ as the corresponding author is that she helps me a lot in reviewing the manuscript and providing the funding for polishing the manuscript. Comments 3: [It is known that feline panleukopenia viruses as well as canine parvovirus type 2 variants can cause infection in cats. Therefore, the limitations of the article should be explained. The success of this VLP vaccine, which was prepared only for a variant in China, in infections against other strains should not be questioned. In this context, the findings obtained are specific only for this variant.] Response 3: Dear reviewer, thank you for pointing this out. Indeed, canine parvovirus type 2 variants (CPV-2a, 2b and 2c) both can cause infection in cats, but it is uncommon (<10% of cause, doi: 10.1016/j.cvsm.2019.02.006), the major pathogen of feline panleukopenia is feline panleukopenia virus. So, the direction of cat vaccine development remains focus on feline panleukopenia virus. The current research is based on an epidemic stain of feline panleukopenia virus that widely circled in China since 2019, so the first consideration is whether the VLP-based vaccine has immunological protection from the FPLV variant. But it not means that the VLP vaccine just prepared only for the Ala91Ser FPLV variant. As to the immune protection effect of VLP vaccine against other FPLV strains will be studied in next step, if I can collected the other FPLV strain, to enrich our vaccine research data. Comments 4: [It is important to include in the discussion how many cats are affected by this new variant. The reason that led researchers to prepare a vaccine against this new variant should be explained.] Response 4: Dear reviewer, thank you for pointing this out. Following your suggestion, two paragraphs were added to the section discussion to provide some information about FPLV variant (Ala91Ser), how many cats are affected by this new variant, and the reason that led author to prepare vaccine against FPLV. As to why use FPLV variant to challenge the immunized cats is that the current research is based on FPLV variant, so the first consideration is whether the VLP-based vaccine has immunological protection from the FPLV variant. As to the immune protection effect of VLP vaccine against previous FPLV strains will be studied in next step, to enrich our vaccine research data. Details can be seen in page 11, ling 289-305. Comments 5: [In Figure 6, only the situation in the muscles at the injection site was evaluated. Why was necropsy not performed on other tissues? Or if it was performed, why were the results not shared? It is important to minimize the effects of Parvoviruses, which cause active infection in the gastrointestinal tract, on this system with vaccination. In particular, the immunohistochemistry results between the unvaccinated group and the vaccinated group that received oral challenge virus are important to provide information about the success of this vaccine. It is recommended that this information be added to the discussion.] Response 5: Dear reviewer, thank you for pointing this out. The purpose of the figure 6 is to evaluate the safety of the vaccine, neither to evaluate the effect of parvovirus on gastrointestinal tract, nor to evaluate the effect of vaccine on reducing the virus replication in gastrointestinal tract. Choose muscle tissue in inoculate site of vaccine was used to check for sarcoma or necrotic lesions that formed by vaccination. So, we not perform necropsy on other tissue and immunohistochemistry analysis between in unvaccinated and vaccinated group. Comments 6: [Figure 2B is an unclear image. The kilodalton size of the VLP bands is not specified?] Response 6: Dear reviewer, thank you for pointing this out. Following your suggestion, we have replaced figure 2B with a new clear figure. Details can be seen in page 8, Figure 2, line 265-274. |
Special thanks to you for your good comments.

Reviewer 2 Report
Comments and Suggestions for Authors
The manuscript by Erkai Feng et al. examines a feline panleukopenia virus (FPLV) virus-like particle (VLP) vaccine, focusing on an endemic strain (Ala91Ser/Ile101Thr mutant) in China, to address the lack of protection of existing vaccines and to fill the gaps in regional epidemic prevention and control. Emphasizing the high mortality rate of FPL (up to 50-90% in young cats) and the increasing rate of breakthrough infections (27%) after vaccination in China highlights the urgency and public health significance of the study. The study covers the whole process of vaccine development from gene cloning, VLPs generation, purification, immunogenicity testing to animal experiments, with rigorous and logical experimental design.
Major issues:
1. The manuscript only observes changes in antibody titers within 50 days of immunization and does not assess long-term protection. This limits the understanding of the long-term effectiveness of the vaccine, especially its ability to provide durable protection in practical applications. It is recommended that antibody titers be tested at time points such as 6 months and 1 year after immunization, and that viral challenge experiments be performed to assess long-term protective effects. Multiple time points (e.g., 3 months, 6 months, 1 year) can be set for antibody titer testing and viral challenge experiments can be performed after each time point. Changes in antibody titers at different time points are analyzed and evaluated for statistical significance.
The manuscript was challenged using only the popular Chinese FPLV variant (Ala91Ser/Ile101Thr) and did not test the cross-protective effect of the VLPs vaccine against other strains. If other FPLV strains (e.g., classical strains or internationally popular strains) are selected for cross-protection experiments. Challenging with different strains and evaluating the cross-protection effect of VLPs vaccines allows for a more comprehensive assessment of the long-term protective effect and broad-spectrum protective capacity of VLPs vaccines. This not only enhances the scientific validity and credibility of the study, but also provides stronger support for the practical application of the vaccine.
In addition, although the background, prevalence, and available research on feline microviruses are not adequately described in the introductory section of the manuscript, it is recommended that the introductory section be expanded to provide more comprehensive background information.
Minor issues:
1.Line 20: Feline panleukopenia (FPL), caused by the feline panleukopenia virus (FPLV), is a severe and highly contagious viral disease with high morbidity and mortality. Added "the" before "feline panleukopenia virus" for clarity and consistency.
2.Line 21: Vaccination remains the gold standard for preventing and controlling this debilitating condition. Corrected the verb form "prevention" and added "of" after "control" for grammatical accuracy.
3.Line 23: The viral protein (VP2) serves as the major immunogen of FPLV and represents the key target antigen in the development of a novel FPLV vaccine. Capitalized "vaccine," added "a" before "novel," and corrected "Vaccine" to "vaccine" for consistency and accuracy.
4.Line 57-60: The emergence of virus-like particles (VLPs) has revolutionized vaccinology by integrating structural virology with precise immunology. Unlike traditional vaccines, VLPs leverage the evolutionary sophistication of viral architecture while mitigating replicative risks, positioning them as a groundbreaking technology for both prophylactic and therapeutic applications. Clarified and enhanced the description for better clarity and flow.
5.Line 81: These findings establish FPLV-VLPs as a cornerstone in pan-parvovirus vaccine development, with translational potential expanding to canine parvovirus (CPV) and porcine parvovirus (PPV). Changed "position" to "establish" for stronger assertion and clarity. Improved sentence structure for coherence.
6.Line 310: FPLV-VP2-VLPs exhibit strong immunogenicity, triggering a notable humoral immune response in vaccinated cats. Replaced "demonstrate" with "exhibit" for variety and clarity. Reworded for better flow and comprehension.
7.Line 318: Conversely, cats given the commercial Fel-O-Vax® PCT vaccine exhibited delayed antibody detection, with HI titers staying below 28 until 21 dpv (Fig. 4, blue line/scatter). Provided contrast using "Conversely," corrected tense consistency, and improved sentence structure for coherence.
Author Response
Response to Reviewer 2 Comments
|
||
1. Summary |
|
|
Thank you very much for taking the time to review this manuscript” Generation and immunogenicity of virus-like particles based on Capsid Protein of a Chinese Epidemic Strain of Feline Panleukopenia Virus (vetsci-3574053)”. Those comments are all valuable and very helpful for revising and improving my paper. We have studied comments carefully and have made correction which we hope meet with approval. Please find the detailed responses below and the corresponding corrections highlighted in pink in the re-submitted files. |
||
2. Point-by-point response to Comments and Suggestions for Authors Major issues: |
||
Comments 1: [The manuscript only observes changes in antibody titers within 50 days of immunization and does not assess long-term protection. This limits the understanding of the long-term effectiveness of the vaccine, especially its ability to provide durable protection in practical applications. It is recommended that antibody titers be tested at time points such as 6 months and 1 year after immunization, and that viral challenge experiments be performed to assess long-term protective effects. Multiple time points (e.g., 3 months, 6 months, 1 year) can be set for antibody titer testing and viral challenge experiments can be performed after each time point. Changes in antibody titers at different time points are analyzed and evaluated for statistical significance.] |
||
Response 1: Dear reviewer, thank you for pointing this out. This study is the basic research process of vaccine development, so we just study the ability of vaccine to induce the hosts to produce protective antibody, and the immune-protective of vaccine against challenged with the virulent FPLV variant, so we challenged the all cats with the virulent FPLV strain at 21 days post vaccination. The reason why observes changes in antibody titers within 50 days post immunization is that there were no deaths occurred in the cats in group I, and II, and we want to know the changes of the cats’ antibodies after challenge, so, we observe changes in antibodies titers within 50 days after immunization. |
||
Comments 2: [The manuscript was challenged using only the popular Chinese FPLV variant (Ala91Ser/Ile101Thr) and did not test the cross-protective effect of the VLPs vaccine against other strains. If other FPLV strains (e.g., classical strains or internationally popular strains) are selected for cross-protection experiments. Challenging with different strains and evaluating the cross-protection effect of VLPs vaccines allows for a more comprehensive assessment of the long-term protective effect and broad-spectrum protective capacity of VLPs vaccines. This not only enhances the scientific validity and credibility of the study, but also provides stronger support for the practical application of the vaccine.] |
||
Response 2: Dear reviewer, thank you for pointing this out. The current research is based on the isolated FPLV variant (Ala91Ser/Ile101Thr), so when we are ready to evaluate the effectiveness of the VLP-based vaccine against FPLV infection, the first choice is our own isolated FPLV variant. Moreover, there were no other FPLV strains in our laboratory. Above all, there was not define of classical strains or internationally popular strains in feline virus research, not like the pig virus. The second reason is that the virus strain used to challenge-protection experiment must have independent intellectual property rights. Beyond that, to my knowledge, there were no literature reported on the cross-protective effect of vaccine against different FPLV strains during the vaccine development so far. The reason may be associated with the fact that there were no significant differences observed among the FPLV isolates, in contrast to the canine parvovirus type 2 variants (CPV-2, CPV-2a, 2b, 2c, new CPV-2a, and new CPV-2b). Comments 3: [In addition, although the background, prevalence, and available research on feline microviruses are not adequately described in the introductory section of the manuscript, it is recommended that the introductory section be expanded to provide more comprehensive background information.] Response 3: Dear reviewer, thank you for pointing this out. Following your suggestion, we re-write the section of introduction, and provide some information about the characteristic of feline panleukopenia virus, the status quo of cat vaccine in China, and why I carry out this research. Details can be seen in page 1-2, section introduction, line 39-55. In addition, we also provide some information about the FPLV variant to the section discussion. Details can be seen in page 10, section discussion, line 295-300. Minor issues: Comments 1: [Line 20: Feline panleukopenia (FPL), caused by the feline panleukopenia virus (FPLV), is a severe and highly contagious viral disease with high morbidity and mortality. Added "the" before "feline panleukopenia virus" for clarity and consistency.] Response 1: Dear reviewer, thank you for pointing this out. Following your suggestion, we have added definite article” the” before “feline panleukopenia virus” for clarity and consistency. Details were highlighted in pink in page 1, section abstract, line 20, in manuscript-revised. Comments 2: [Line 21: Vaccination remains the gold standard for preventing and controlling this debilitating condition. Corrected the verb form "prevention" and added "of" after "control" for grammatical accuracy.] Response 2: The noun (n.) ‘prevention’ was changed into verb (v.) ‘preventing’, and the preposition(prep.) ’of ’was added to the behind of ‘control’. Dear reviewer, agree with your comment. We have changed the word class of ‘prevent’ from the noun (n.)’prevention’ to the verb (v.) ‘preventing’ for grammatical accuracy. In addition, we have added the preposition (prep.) ‘of’ to the behind of the verb (v.) ‘control’ for grammatical accuracy. Details were highlighted in pink in page 1, section abstract, line 22. |
Comments 3: [Line 23: The viral protein (VP2) serves as the major immunogen of FPLV and represents the key target antigen in the development of a novel FPLV vaccine. Capitalized "vaccine," added "a" before "novel," and corrected "Vaccine" to "vaccine" for consistency and accuracy.]
Response 3: Dear reviewer, thank you for pointing this out. Following your suggestion, we had changed the capitalized letter ‘V’ in word ‘Vaccine’ into lowercase ’v’, and corrected “Vaccine” to ‘vaccine’. In addition, we added indefinite article ‘a’ before the adjective ’novel’. Details can be seen in page 1, line 24.
Special thanks to you for your good comments.

Reviewer 3 Report
Comments and Suggestions for Authors
The manuscript titled “Generation and immunogenicity of virus-like particles based 2 on Capsid Protein of a Chinese Epidemic Strain of Feline 3 Panleukopenia Virus” by Erkai et al., describes the generation of VLPs containing Feline Panleukopenia virus capsid protein VP2 and asses its immunogenicity. The authors demonstrate that the immunization with these VLPs elicit strong immune response as it provides complete protection against FPLV infection in cats. The authors are advised to address following concerns:
1) Line 16 in Abstract mentioned “strong IgG-specific antibody response”. However, there is no experiment in the manuscript to define the fraction of antibodies produced post immunization. Please avoid using such terms without providing experimental evidences.
2) Please include (in supplementary data) vector maps of pFastBac-260 FPLV-VP2 and rBacmid-FPLV-VP2 in support of gel pictures shown in Figure-1. It is difficult to understand and verify cloning strategy without a proper vector map or graphic design of constructs.
3) Western blot image in Figure-2B shows multiple bands in addition with 70 kDa FPLV-VP2-VLP protein. Does this mean that the protein is multimerized or degraded while purification? It will be nice to show a clean WB image with single band of FPLV-VP2-VLP protein. It will confirm that the immunogen is a single purified peptide.
4) Author’s conclusion about FPLV-VP2 protein expression in Figure-3 is inconclusive. IF based GFP signal detected in the infected cells does not mean that they are expressing FPLV-VP2 protein correctly. Only a WB analysis of cell extracts from infected cells could confirm if the FPLV-VP2 protein is expressed correctly. Please show a WB analysis of cell lysate prepared from these infected cells.
5) There is no description of how the HE staining was performed as shown in Figure-6. Please give details of HE staining in Method section.
6) Authors show FPLV-VLPs vaccine gave 100% protection again FVLP infection as compared to commercial or killed vaccine. However, there is no experimental evidence included in the manuscript which could confirm reduced virus replication in these cats. Please provide some additional data such as qPCR or IHC analysis in support of abrogation of virus replication and cure from FVLP infection in the immunized animals.
7) The manuscript needs a thorough check for inconstancy of used terms, flow and grammatical correction. Following are some examples:
- Line 343; please check if this line is correct and fits to the context.
- Line 212; replace the word “seriously” with “serially”.
Author Response
Response to Reviewer 3 Comments |
||
1. Summary |
|
|
Thank you very much for taking the time to review this manuscript” Generation and immunogenicity of virus-like particles based on Capsid Protein of a Chinese Epidemic Strain of Feline 3 Panleukopenia Virus (vetsci-3574053)”. Those comments are all valuable and very helpful for revising and improving my paper. We have studied comments carefully and have made correction which we hope meet with approval. Please find the detailed responses below and the corresponding corrections highlighted in yellow in the re-submitted files. |
||
2. Point-by-point response to Comments and Suggestions for Authors |
||
Comments 1: [ Line 16 in Abstract mentioned “strong IgG-specific antibody response”. However, there is no experiment in the manuscript to define the fraction of antibodies produced post immunization. Please avoid using such terms without providing experimental evidences.] |
||
Response 1: [deleted ‘IgG-’ from the line 16 in Abstract and mark the sentence ’specific antibody response’ in red] Dear reviewer, thank you for pointing this out and We agree with this comment. Therefore, I have deleted the words ‘IgG-’ from line 16 in Abstract. Detailed can be seen in page 1, paragraph Simple Summary, line 16. |
||
Comments 2: [Please include (in supplementary data) vector maps of pFastBac-260 FPLV-VP2 and rBacmid-FPLV-VP2 in support of gel pictures shown in Figure-1. It is difficult to understand and verify cloning strategy without a proper vector map or graphic design of constructs.] |
||
Response 2: Dear reviewer, agree with your comment and I have supplemented two vector maps of pFastBac-FPLV-VP2 and rBacmid-FPLV-VP2 in supplementary data, which was named pFastBac-FPLV-VP2, and rBacmid-FPLV-VP2, respectively. In addition, a flow chart map of construction of pFastBac-FPLV-VP2 was added to supplementary data, named flow chart of construction of pFastBac-FPLV-VP2. |
||
Comments 3: [Western blot image in Figure-2B shows multiple bands in addition with 70 kDa FPLV-VP2-VLP protein. Does this mean that the protein is multimerized or degraded while purification? It will be nice to show a clean WB image with single band of FPLV-VP2-VLP protein. It will confirm that the immunogen is a single purified peptide.] Response 3: Dear reviewer, thank you for pointing this out. In this study, FPLV-VP2-VLP is a crude extract from the Sf9 cell culture after treated with 25 mM NaHCO3, degradation is inevitable. If want to show a clean WB image with a single band will be easily achieved through sucrose density gradient centrifugation, but it may greatly increase the manufacturing cost of VLP-base vaccine. In addition, the prepared VLPs vaccine base on the crude extract from the Sf9 cells in this study showed higher safety to cats, and no adverse side reaction in inoculation site. All in all, following your suggestion, I will provide a clean WB image with a single band of FPLV-VP2-VLP. Details can be seen in Comments 4: [Author’s conclusion about FPLV-VP2 protein expression in Figure-3 is inconclusive. IF based GFP signal detected in the infected cells does not mean that they are expressing FPLV-VP2 protein correctly. Only a WB analysis of cell extracts from infected cells could confirm if the FPLV-VP2 protein is expressed correctly. Please show a WB analysis of cell lysate prepared from these infected cells.] Response 4: Dear reviewer, thank you for pointing this out. Agree with your comment, detected GFP signal just mean the VP2 protein was expressed in Sf9 cells and could be specifically recognized by the monoclonal antibody. In this study. we have showed a WB analysis of cell lysate prepared from these infected cells in Figure-2B. Comments 5: [There is no description of how the HE staining was performed as shown in Figure-6. Please give details of HE staining in Method section.] Response 5: Dear reviewer, thank you for pointing this out and agree with your comments. We have supplemented description of how the sampling and HE staining in Method section. Details can be seen in page 5, Method section, line 187-193. Comments 6: [ Authors show FPLV-VLPs vaccine gave 100% protection again FVLP infection as compared to commercial or killed vaccine. However, there is no experimental evidence included in the manuscript which could confirm reduced virus replication in these cats. Please provide some additional data such as qPCR or IHC analysis in support of abrogation of virus replication and cure from FVLP infection in the immunized animals.] Response 6: Dear reviewer, thank you for pointing this out. However, we did not conduct research on the vaccine effect in abrogation of virus replication or cure from FPLV infection in the immunized animals. Generally speaking, researchers seldom pay attention to the effect of vaccine in abrogating the virus replication or curing from FPLV infection during the economic animal or pet vaccine development. The focus is more on the titer of specific antibody that induced by the vaccine, the reduction of clinical symptoms, and challenge-protection rate the immunized animals. 4. Response to Comments on the Quality of English Language |
||
Point 1: The manuscript needs a thorough check for inconstancy of used terms, flow and grammatical correction. Following are some examples:1. Line 343; please check if this line is correct and fits to the context. 2. Line 212; replace the word “seriously” with “serially”. |
||
Response 1: Dear reviewer, thank you for pointing this out. Following your suggestion, we have checked for inconstancy of used terms, slow and grammatical correction. For example, we re-write the line 343” Collectively, these finding demonstrates the higher safety profile of VLPs-based vac-cine”, details can be seen in page 12, line 342-343. In addition, we also changed the word” seriously” into “serially”, details can be seen in page 5, line 218. |
||
|
||
Special thanks to you for your good comments.
|

Round 2
Reviewer 1 Report
Comments and Suggestions for Authors
Researchers have made some of the expected corrections, but some issues still need to be clarified.
The turnitin report showing that the similarity rate dropped to 31% is attached. Here, number 1 is the work that does not belong to the current article authors and the similarity was found to be high at 9%.
Number 1 article: Arch Virol. 2020 Sep;165(9):2065-2071. doi: 10.1007/s00705-020-04703-6. Epub 2020 Jul 1: Generation and immunogenicity of virus-like particles based on mink enteritis virus capsid protein VP2 expressed in Sf9 cells.
It is recommended that the disclaimers from the 5 people included in the previous article version but not in the mdpi version be forwarded to the journal management as a separate file. These documents are important against possible legal objections in the future.
It is recommended that the statements in Response 3 be written as a future plan in the discussion section of the article.
What is the VLP size in kDa? Although it is stated in the results section (3.2 Analysis of FPLV-VP2-VLPs) that the protein is 70 kDA in size, it is seen that it is below the 70 kDa marker in line 2 in Figure 1 A.
While all other figures are included in the text, Fig 2B is not used in the text.
Legend of figure 6 A should be corrected as necropsy not “necrosy”.

Legend of figure 6 A should be corrected as necropsy not “necrosy”.
Author Response
Response to Reviewer 1 Comments |
||
1. Summary |
|
|
Thank you very much for taking the time to review this manuscript” Generation and immunogenicity of virus-like particles based on Capsid Protein of a Chinese Epidemic Strain of Feline Panleukopenia Virus (vetsci-3574053)”. Those comments are all valuable and very helpful for revising and improving my paper. We have studied comments carefully and have made correction which we hope meet with approval. Please find the detailed responses below and the corresponding corrections highlighted in yellow in the re-submitted files. |
||
2. Point-by-point response to Comments and Suggestions for Authors |
||
Comments 1: [The turn it in report showing that the similarity rate dropped to 31% is attached. Here, number 1 is the work that does not belong to the current article authors and the similarity was found to be high at 9%.] |
||
Response 1: Dear reviewer, thank you for pointing this out again, but I do not agree with this comment. In the last response to reviewer, I once doubt that the checked two articles that show the highest similarity (9%, 8%) to the vetsci-3574053 is a pre-printed version of my manuscript that once submitted to a journal, whose name is “Baculovirus Expression of Feline Panleukopenia Virus Capsid Protein and Induces Robust Immune Responses in Cats”. However, your response to the doubt is tell me the article that show highest similarity to the vetsci-3574053 is an article that titled “Generation and immunogenicity of virus-like particles based on mink enteritis virus capsid protein VP2 expressed in Sf9 cells”. After that, I immediately checked the referred article, but I do not find the higher similarity between the vetsci-3574053 and that article. In contrast, we found much more similarity that label number 1 (superscript) and number 2 (superscript) in revision document (vetsci-3574053-peer-review-v2_1.pdf) were found in the pre-printed manuscript of mine, like “by Nanjing Zoonbio Biotech Co., Ltd.(Nanjing)” in line 83, “anti-CPV-2c-VP2 monoclonal antibody(clone 5B1B)”in line 79, “(Rotator type F-34-6-38, Eppendorf, Germany)”in line 126, “the antigenicity of the FPLV-VP2-VLPs protein(designed as P4) was evaluated using rapid test immunochromatographic strips developed by SHANGHAI QUICKING Biotech CO., Ltd. “ in line 133-135, and so on. All these text description were either associated with the biotech company that I chose to optimized and synthesized the full-length of VP2 gene, or related to the instrument (centrifuge of Eppendorf) and homemade monoclonal antibody. In addition, some legends or the title of papers are also showing similarity with the check articles. So, I firmly believe that the article that showed highest similarity to vetsci-3574053 is my pre-submitted manuscript (Baculovirus Expression of Feline Panleukopenia Virus Capsid Protein and Induces Robust Immune Responses in Cats), rather than the article entitle ”Generation and immunogenicity of virus-like particles based on mink enteritis virus capsid protein VP2 expressed in Sf9 cells”. We sincerely hope that the reviewer can offer their comments. No matter how it turns out, we have rewrite the similar sentences in revision document (vetsci-3574053-peer-review-v2_1.pdf), so as to can drop the similarity between the vetsci-3574053 and the checked articles. All re-write sentences in manuscript-revised were recorded in yellow. |
||
Comments 2: [It is recommended that the disclaimers from the 5 people included in the previous article version but not in the mdpi version be forwarded to the journal management as a separate file. These documents are important against possible legal objections in the future.] |
||
Response 2: Dear reviewer, thank you for pointing this out. Following your suggestion, I have gained the disclaimers from the 5 people (Manping Yan, Jinyuan Shang, Zhenjun Wang,Shipeng Cheng and Xiaohao Zhang) included in the previous article. Details can be seen in Supplement materials. |
||
Comments 3: [What is the VLP size in kDa? Although it is stated in the results section (3.2 Analysis of FPLV-VP2-VLPs) that the protein is 70 kDA in size, it is seen that it is below the 70 kDa marker in line 2 in Figure 1 A.] |
||
Response 3: Dear reviewer, thank you for pointing this out. The deduced size of VLPs is 70683.4 daltons. As to the VLPs is seen below the 70 kDa marker may be result from the treatments on picture, like stretching, cropping, and compositing. |
||
Comments 4: [While all other figures are included in the text, Fig 2B is not used in the text.] |
||
Response 4: Dear reviewer, thank you for pointing this out. The fig.2B is a new WB picture that used purified FPLV-VLPs that treated with the ultrafiltration centrifuge tube (50 kDa, Millipore, Germany), for you reminded me to provide a clearer and single-component WB image. But supplied this step in preparing the VLPs and VLPs-based vaccine, it will increase the cost of VLPs-based vaccine, for the price of ultrafiltration centrifuge tube is not cheap. |
||
Comments 5 : [Legend of figure 6 A should be corrected as necropsy not “necrosy”.] |
||
Response 5: Dear reviewer, thank you for pointing this out. Following your suggestion, we have changed the wrong word ’necrosy’ with ‘necropsy’ in the legend of figure 6. details can be seen in the new figure 6 that in the manuscript-revised, in page 11, line 294. |
Special thanks to you for your good comments.

Reviewer 3 Report
Comments and Suggestions for Authors
Thanks for sharing the vector maps for the better understanding of cloning strategy. I have few minor comments:
1) The term “IgG” was removed, however using word “specific” is confusing here. Please correct it.
2) Authors are appreciated to include a clean WB image for FPLV-VP2-VLP protein. However, author's reasoning that purifying this protein from from fs9 crude extract “may greatly increase the manufacturing cost of VLP-based vaccine” is not justified. The sf9 crude is a pool of several unidentified cellular proteins along with the large amount of FPLV-VP2-VLP protein. Authors are suggested to cross check if they have used crude extract instead of purified protein for vaccination and mention the same in the manuscript.
The authors have substantially addressed the suggested corrections therefore I do not have any further concerns.
Author Response
Response to Reviewer 3 Comments |
||
1. Summary |
|
|
Thank you very much for taking the time to review this manuscript” Generation and immunogenicity of virus-like particles based on Capsid Protein of a Chinese Epidemic Strain of Feline Panleukopenia Virus (vetsci-3574053)”. Those comments are all valuable and very helpful for revising and improving my paper. We have studied comments carefully and have made correction which we hope meet with approval. Please find the detailed responses below and the corresponding corrections highlighted in gree in the re-submitted files. |
||
2. Point-by-point response to Comments and Suggestions for Authors |
||
Comments 1: [The term “IgG” was removed, however using word “specific” is confusing here. Please correct it.] |
||
Response 1: Dear reviewer, thank you for pointing this out. Following your suggestion, we have deleted the word ’specific’ from the sentence ’Immunization with this vaccine elicited a strong specific antibody response’ in line 15-16, page 1, and the sentence ’which were identified using a monoclonal antibody specific to the expressed VP2’ in line 317, page 11. Details can be seen in page 1, line 16, and page 11, line 317, in the manuscript-revised. |
||
Comments 2: [Authors are appreciated to include a clean WB image for FPLV-VP2-VLP protein. However, author's reasoning that purifying this protein from fs9 crude extract “may greatly increase the manufacturing cost of VLP-based vaccine” is not justified. The sf9 crude is a pool of several unidentified cellular proteins along with the large amount of FPLV-VP2-VLP protein. Authors are suggested to cross check if they have used crude extract instead of purified protein for vaccination and mention the same in the manuscript.] |
||
Response 2: Dear reviewer, thank you for pointing this out. In my opinion, there were no need to purify the FPLV-VLPs crude, because there was no any adverse reaction happen on cats, like fever, and no ulcer or no sarcomas were detected at the injection site. So, we speculated the several unidentified proteins maybe the degradation products of FPLV-VLPs, and further think it maybe not need purified, for purifying proteins either by centrifugation (ultrafiltration centrifuge tube), or by sucrose density gradient centrifugation, both all very expensive. In terms of production scale, the cost of purifying will be higher. In addition, added more steps in prepare the VLPs-based vaccine may be also increased the cost of vaccine. As to the reviewer’s suggestion to compare the crude extract and purified protein for vaccination, I think there was no need to do, as everyone knows, the purified protein will be better than the crude. However, compare to the higher cost, whether it is good enough to do is debatable. In this study, there were no side effect was found in cats that associated with vaccination with the crude FPLV-VLPs, So, this is why we decided not to purify the crude extracted FPLV-VLPs. |
Special thanks to you for your good comments.
